# Comparison between Dry-Land and Swimming Priming on 50 m Crawl Performance in Well-Trained Adolescent Swimmers

**DOI:** 10.3390/sports10040052

**Published:** 2022-03-31

**Authors:** Nikolaos Zaras, Andreas Apostolidis, Angeliki Kavvoura, Marios Hadjicharalambous

**Affiliations:** 1Human Performance Laboratory, Department of Life and Health Sciences, University of Nicosia, 46 Makedonitissas Ave., P.O. Box 24005, Nicosia 1700, Cyprus; apostolidis.a@unic.ac.cy (A.A.); hadjiharalambous.m@unic.ac.cy (M.H.); 2School of Physical Education and Sports Science, National and Kapodistrian University of Athens, Ethnikis Antistassis 41, 172 37 Daphne, Greece; a_kavvoura@hotmail.com

**Keywords:** priming, swimming, crawl, rate of torque development, time-trial

## Abstract

The purpose of the study was to investigate the effect of dry-land priming (DLP) versus swimming priming (SP) on the 50 m crawl performance of well-trained adolescent swimmers. Thirteen adolescent swimmers were randomly assigned to perform either a DLP or SP 24 h prior to a 50 m sprint crawl time-trial. Baseline measurements included a 50 m sprint crawl time-trial as a control (C) condition, the evaluation of body composition, countermovement jump (CMJ), isometric peak torque (IPT), and rate of torque development (RTD). Rating of perceived exertion (RPE) was obtained following the DLP and SP programs. Both DLP and SP significantly decreased the 50 m crawl time-trial, by −2.51 ± 2.43% and −2.59 ± 1.89% (*p* < 0.01), respectively, compared with the C time-trial. RPE was not different between DLP and SP (*p* = 0.919). CMJ performance remained unchanged after DLP and SP programs compared with the C trial (*p* > 0.05). The percentage decrease in the 50 m crawl after DLP was significantly correlated with the percentage decrease in the 50 m crawl following SP (r = 0.720, *p* = 0.006). CMJ power, lean body mass, IPT, and RTD were significantly correlated with 50 m crawl performance. These results suggest that both DLP and SP strategies, when applied 24 h prior to a 50 m crawl time-trial, may enhance performance in well-trained adolescent swimmers.

## 1. Introduction

Priming is a training strategy performed primarily 6 to 33 h before the main competition aiming to enhance the athlete’s preparedness and performance [1]. Priming training sessions are short in duration (approximately 30 min) and may focus on the enhancement of strength, speed, and/or specific motor skills of various sports, such as ball dribble in soccer and free shots in basketball [2]. Indeed, studies have shown that a priming training session 5–6 h before competition may enhance performance in team-sport athletes mainly by maintaining the testosterone levels in the afternoon competition [3,4,5], as well as by enhancing neuromuscular activation and the rate of force development (RFD) for the upcoming event [6,7]. Increased core temperature has also been proposed as an important factor for an elevated performance by increasing the metabolic rate during the afternoon performance task [8,9], while priming, similar to post activation performance enhancement (PAPE), may increase the muscle fiber sensitivity to calcium ions (Ca^2+^), leading to an increased muscle contraction activation and subsequently an increased performance [10,11]. Moreover, increased mechanical stiffness has also been proposed as a potential mechanism of increased performance due to the positive correlation observed between joint stiffness and neuromuscular performance [6,12]. However, the primary mechanism of increased performance following priming training remains unclear.

Dry-land training and resistance exercise are key factors for performance enhancement in sprint swimming [13,14,15], with studies suggesting that for gaining the best benefit of resistance training, this should be performed with maximum intentional movement velocity [13,16,17]. However, little is known concerning the role of dry-land priming in swimming. A study in national level swimmers showed that a morning priming session (6 h before the evening main time-trial) consisting of either swimming only or a combination of dry-land resistance exercise and swimming may significantly enhance performance in 100 m sprint swimming by 1.6 ± 0.6% and 1.7 ± 0.7%, respectively, compared to a control condition, without observing any significant difference between experimental conditions [8]. In addition, a more recent study in national and international swimmers revealed that an ischemic preconditioning training strategy, 2 and 24 h before 100 m and 200 m time trial swimming, induced no significant changes in time trial performance [18]. An important question that concerns coaches and athletes is which might be the best pre-competition training strategy in order to increase sprint swim performance. Consequently, whether a power based priming or a high-intensity swimming priming training sessions, when applied 24 h prior to competition, may increase performance in sprint swimming, remains to be elucidated. 

Dry-land training may enhance muscular strength leading to increase in lean body mass [19,20,21]. A previous study in adolescent swimmers showed a significant correlation between lean body mass and 100 m freestyle swimming performance (r = −0.26, N = 280) with faster swimmers possessing greater lean mass compared to lower performance swimmers [22]. In addition, a subsequent study showed that muscularity may predict 100 m swimming performance in boys and girls [23]. These correlational results imply that increases in body composition and lean mass may result in significant enhancement in sprint swimming performance. However, the available correlation analysis data between lean body mass and sprint swimming are scarce. In addition, although several studies have shown significant correlations between maximum strength and swimming performance [20,24,25], rare data exist regarding the relationship between RFD and sprint swimming performance. Loturco et al. [20] found a significant correlation between RFD and 50 m sprint (r = −0.72) in a group of well-trained young swimmers whereas a study in international swimmers found a non-significant correlation (r = −0.56, *p* > 0.05) between dynamic RFD (calculated from countermovement jump; CMJ) and 15 m sprint swim [26]. Hence, the correlation between RFD in specific time windows and sprint swim performance needs further investigation.

The aim of the present study was therefore twofold: (a) to examine the effect of a power-focused dry-land priming session versus a swimming priming session performed 24 h before the time-trial in 50 m crawl swimming and (b) to investigate the relationships between body composition, power, and RFD in a group of well-trained adolescent swimmers. The hypotheses of the study were: (a) both priming training sessions will enhance swimming performance and (b) lean body mass, power, and RFD will strongly be correlated with time-trial performance in 50 m crawl.

## 2. Methods

### 2.1. Participants

Thirteen well-trained adolescent swimmers, 11 males (age: 14.6 ± 0.9 years; body mass: 62.9 ± 8.5 kg; body height: 1.71 ± 0.05 m) and 2 females (age: 15.5 ± 2.1 years, body mass: 59.4 ± 7.6 kg, body height: 1.63 ± 0.03 m), with 9.8 ± 1.2 years of training experience and 5.9 ± 1.6 years of competition experience participated in the study. All athletes were competing in national swimming competitions and their best performance in 50 m crawl amounted to 77.5 ± 7.2% of the best performance in the nation for 2021. Athletes fulfilled the following criteria: absence of any orthopedic/neuromuscular maladies, absence of drug abuse or nutritional supplements, while they should at least have participated in a systematic daily swimming training during the previous 2 years. All athletes participated in crawl swimming events from 50 m to 200 m and followed 5–6 training sessions per week (approximately 2 dry-land training sessions and 3–4 swimming training sessions). Athletes and their parents were informed about the experimental procedures and all signed an informed consent form. All procedures were in accordance with the 1975 Declaration of Helsinki as revised in 2013 and were approved by the National Ethics Committee (project number ΕΕΒΚ/ΕΠ/2020/55; 04/12/2020).

### 2.2. Procedures

Experimental procedures were performed during the specific preparation phase in which the training volume was high and training intensity was medium to high. Athletes completed two different priming training sessions, using a counterbalanced design, during a 3-week experimental period (Figure 1). During the first week, all athletes visited the laboratory on two different days: (a) for the evaluation of body mass and body composition and for performing the familiarization session with power measurements, and (b) for the assessment of CMJ, lower body isometric peak torque (IPT) production, and rate of torque development (RTD). In the same week, following 48 h of rest, athletes performed a maximum 50 m time-trial crawl swim which was considered as the control condition (C). The second week, athletes were randomly assigned into dry-land priming (DLP: N = 6) and swimming priming (SP: N = 7) groups. The opposite assignment was performed during the third week. Consequently, all athletes completed all three conditions: C, DLP, and SP. Prior to the time-trials, all athletes were weighed on the same portable body scale. In addition, 30 min after the 50 m crawl time-trials, athletes performed 3 maximum CMJ trials. Changes in 50 m crawl swimming, body mass, and CMJ were compared between C, DLP, and SP, while a correlational analysis was used to explore the possible relationships between all variables.

### 2.3. Training Intervention

All athletes followed a similar training program with small individual differences under the supervision of the same qualified swimming coach. Priming training was performed after a rest day. Athletes were instructed to be well fed and hydrated before both priming sessions as well as before the 50 m crawl time evaluation. Table 1 presents the acute priming training programs for DLP and SP. For DLP, athletes followed a 10-min warm-up including static stretching exercises for all muscle groups, dynamic exercises such as static skipping, and powerful movements of the lower and upper body. Power training was then performed using medicine balls, stretch cords, and CMJs [14,19,27]. Athletes were instructed to perform all repetitions with maximum intentional movement velocity [16]. At the end of DLP, athletes finished the training session with static stretching. In addition, SP training was performed completely in the swimming pool. Athletes followed a warm-up that consisted of 200 m crawl and 200 m mixed swimming (50 m of each four styles), as well as 4 × 50 m lower and upper body swimming exercises (15 m streamline kick crawl only, 20 m drill, and 15 m swim crawl) and 100 m crawl swim followed by SP training (Table 1). During SP, athletes were instructed to swim as fast as possible. At the end of SP training, a 200 m crawl swimming for cooling down was performed. Thirty minutes after the priming sessions, athletes provided their rate of perceived exertion (RPE) (Borg scale 6–20) [28]. The duration of the priming training sessions was approximately 30 min.

### 2.4. 50 m Crawl Time Measurement

Performance in 50 m crawl was evaluated in an indoor swimming pool (water temperature 26.7 °C, air temperature 28 °C) under three different conditions: after a rest day (C), after DLP training, and after SP training. Prior to all performance measurements, athletes performed the same warm-up swimming protocol including a 200 m crawl, followed by a 200 m mixed-styles swimming, 4 × 50 m lower and upper body swimming exercises, and another 100 m crawl with increased velocity. Following a 15-min period of passive rest, imitating the competition preparation routine, athletes performed two maximum 50 m crawl swimming attempts, with a 10-min passive rest between them. During all conditions athletes competed in groups of two to increase the competitive spirit. Overall times were manually measured (Casio, Model HS-80TW-1EF, Mainland UK) by two certified swimming coaches, of which one was a member of the national coaching team. The lower time-trial performance was used for the statistical analysis. The intra-class correlation coefficient (ICC) for 50 m crawl time measurement was 0.96 (95% confident intervals (CI): Lower = 0.86, Upper = 0.98).

### 2.5. Body Composition Analysis

During the first day of the laboratory measurements, body mass and body composition analysis via bioelectrical impedance scale was performed. Measurements were performed during morning hours. Athletes were instructed to fast for approximately 10 h prior to the body composition analysis and to refrain from any strenuous exercise for 24 h [29]. Athletes were initially weighed only with their pool swimsuits on a portable scale (Tanita BC-545n, Southampton, UK) for body mass evaluation. The same scale was used during the days of 50 m time-trials following DLP and SP training. Then, body composition analysis was evaluated (Tanita MC-780MA, Tokyo, Japan) and included body fat, total lean body mass, trunk lean mass, legs lean mass, and arms lean mass. The ICC for body mass, body fat, total lean body mass, trunk lean mass, legs lean mass, and arms lean mass were: 0.998 (95% CI: Lower = 0.998, Upper = 0.999), 0.990 (95% CI: Lower = 0.982, Upper = 0.995), 0.965 (95% CI: Lower = 0.958, Upper = 0.987), 0.965 (95% CI: Lower = 0.986, Upper = 0.994), 0.975 (95% CI: Lower = 0.980, Upper = 0.990), and 0.981 (95% CI: Lower = 0.990, Upper = 0.998), respectively. Following the body composition evaluation, a familiarization session was performed in CMJs and isometric leg extension.

### 2.6. Countermovement Jump

The next day, athletes visited the laboratory for the evaluation of CMJs and lower body isometric force in leg extension. After an 8-min warm-up on a stationary bicycle at 50 Watts and some lower-body dynamic stretching exercises, athletes performed 3 warm-up/familiarization CMJs with lower intensity. Then, athletes performed 5 maximal CMJs with a self-selected depth high (Optojump Modular System, Warwickshire, UK) and with arms akimbo. Between all attempts, 2 min of recovery was allowed. All data from the CMJs were recorded and analyzed (Optojump Next, Warwickshire, UK) to calculate the maximum vertical jump height the power output during the push off phase [30] and power per body mass. From the 5 CMJs, the best jump height performance was used for the statistical analysis. During the two priming sessions, CMJ attempts were performed 30 min after the end of the 50 m crawl time measurements to evaluate any change in power production, potentially induced by the two priming training programs. Athletes followed the same warm-up protocol and then performed 5 maximum CMJs as previously described. The ICC for CMJ height, power, power per body mass were 0.989 (95% CI: Lower = 0.957, Upper = 0.997), 0.980 (95% CI: Lower = 0.985, Upper = 0.990), and 0.981 (95% CI: Lower = 0.978, Upper = 0.991), respectively.

### 2.7. Lower Body Isometric Peak Torque and Rate of Torque Development

Isometric leg extension measurement was performed 15 min following the CMJs on the isokinetic dynamometer (HUMAC NORM isokinetic extremity system, Massachusetts, USA). More specifically, athletes were seated on the isokinetic dynamometer chair and straps were used to ensure a stable body position. The exercising leg was determined during the familiarization session [31] while the knee angle was set at 60° flexion (0° = full extension) as previously described [32]. For the warm-up, three sub-maximal efforts were performed while athletes were instructed to progressively increase their force. Then, 3 maximal efforts were allowed with 2 min rest between attempts and athletes were instructed to apply their maximum force as fast as possible and to sustain it for 3 s [33]. During all attempts, athletes had real-time visual feedback of the force applied via a computer monitor which was placed in front of them. All data collected from the leg extension isometric measurements were recorded and analyzed to calculate the IPT and the RTD from the torque-time curve. IPT was calculated as the greater force generated from the torque-time curve while RTD was calculated as the mean tangential slope of the torque-time curve in specific time windows of 0–20, 0–60, 0–80, 0–100, 0–120, 0–150, 0–200, 0–250, and 0–300 milliseconds. The best performance from the torque-time curve was used for the statistical analysis. The ICC for IPT was 0.990 (95% CI: Lower = 0.964, Upper = 0.998) and the mean for RTD was 0.893 (95% CI: Lower = 0.649, Upper = 0.972).

### 2.8. Statistical Analysis

A prior power analysis was performed for the determination of sample size revealing an actual power of 0.973 for a maximum number of 8 participants for the differences between groups. All values are presented as mean ± SD. All data were normally distributed according to the Kolmogorov–Smirnov test. A 3-way analysis of variance for repeated measures was used to examine differences between C, DLP, and SP. Cohen’s d effect size was also calculated. Paired samples *t*-Test was used to examine RPE differences between DLP and SP. Pearson’s r product moment correlation coefficient was used to explore the relationships between laboratory measurements and performance variables. Reliability of all measurements was performed using a two-way random effect ICC with 95% CI. Significance was accepted at *p* ≤ 0.05.

## 3. Results

All athletes completed all three conditions without injuries. Table 2 presents the results from the laboratory baseline measurements (C) and the results from DLP and SP groups. Performance in the 50 m crawl swim was enhanced significantly following DLP by 2.51 ± 2.43% (*p* = 0.012, d = 0.299) and SP by 2.59 ± 1.89% (*p* = 0.001, d = 0.305), compared to C (Table 2). No significant difference was observed between DLP and SP (*p* = 0.989, d = 0.008) for the 50 m time-trial. RPE was not different between DLP and SP groups (DLP: 6.2 ± 1.9 vs. SP: 6.3 ± 2.8, *p* = 0.919, d = 0.009). No significant changes were observed for body mass, CMJ height, power, or power per body mass following DLP and SP (*p* > 0.05).

A significant correlation was found between the percentage decrease of 50 m crawl time-trial after DLP with the percentage decrease of 50 m crawl time-trial after SP (r = 0.720, *p* = 0.06, N = 13) (Figure 2). Correlation analysis for raw data was performed only for male athletes (N = 11). Significant correlations were found between 50 m crawl time performance following C, DLP, and SP with body composition variables (Table 3). Significant correlations were found only between CMJ power and 50 m crawl time performance after C (r = −0.779, *p* = 0.005), DLP (r = −0.788, *p* = 0.004), and SP (r = −0.749, *p* = 0.008). Furthermore, lower body IPT and RTD were significantly correlated with 50 m crawl time performance following C, DLP, and SP conditions (Table 4).

## 4. Discussion

The main finding of the current study was that performance in 50 m crawl swimming was significantly improved in well-trained adolescent swimmers following both DLP (reduction in time-trial by −2.51%) and SP (reduction in time-trial by −2.59%) when compared to C. Athletes experienced the same RPE following the two priming programs and CMJ performance remained unaltered compared to C. In addition, a significant correlation was found between the percentage decrease in the 50 m crawl time-trial after DLP and SP training strategies, which shows that all athletes experienced similar performance enhancements following both priming programs. Countermovement power production, lean body mass, IPT, and RTD were significantly correlated with 50 m crawl performance, under all experimental conditions. These results suggest that 50 m crawl sprint swimming performance may be significantly enhanced following a power-based dry-land or a swimming priming session applied 24 h before time-trial measurement. Furthermore, based on the current results, CMJ power, lean body mass, and lower body IPT and RTD may be used as simple coaching tools for predicting performance in well-trained adolescent swimmers.

Dry-land training is a common strategy to enhance performance in sprint swimming [14,19,20,21]. An ischemic preconditioning training strategy, 2 and 24 h before 100 m and 200 m time trial swimming, was insufficient to induce significant changes in swimming performance [18]. In contrast, a combination of a morning dry-land and swimming priming versus swimming priming only applied 6 h before a 100 m time-trial was found to enhance performance in national swimmers in comparison to no morning training [8]. The plausible mechanism, given by these authors, for this performance enhancement is that morning priming training elevated body, core, and skin temperature, which may contribute to increasing the metabolic rate that potentially remained elevated until the afternoon main time-trial [8]. In addition, studies in team sports have shown that a morning strength priming session may enhance afternoon performance by offsetting the circadian decline in serum testosterone concentration contributing to enhancing lower-body power output and repeat-sprint performance for up to six hours post-morning priming training [3,4,5]. However, in the current study, it appears unlikely that body temperature and/or hormonal responses could remain elevated 24 h after a priming session [1], while performance in the CMJs as a neuromuscular index [6] remained unchanged following both DLP and SP compared to C. Consequently, the plausible mechanism underling the enhanced performance after DLP and SP 24 h before 50 m crawl time performance needs further investigation.

An interesting finding of the present study was the strong correlation observed between the percentage decrease in 50 m crawl time-trial after DLP and the percentage decrease in 50 m crawl time-trial after SP (r = −0.720), strengthening the conclusion that both training programs may be used effectively by athletes to enhance performance in 50 m crawl. This correlation further reinforces the finding that athletes may experience similar decrements in 50 m crawl time-trial following both priming programs. Hence, from a practical point of view, when athletes have no access to a swimming pool 24 h before competition, coaches may prescribe a dry-land power focused training program in an attempt to enhance next-day swimming performance mainly for sprint swimmers. The results of the current study suggest that athletes may choose either DLP or SP and both may have similar positive effects on 50 m crawl performance.

Correlational analysis revealed significant connections between lean body mass and 50 m crawl performance following C, DLP, and SP conditions. Previous studies have shown that muscularity is a significant contributor in 100 m swimming performance in younger (aged: 10.3 ± 1.0 years) and older (aged: 19.8 ± 1.6 years) swimmers, while a significant correlation was found between lean body mass and 100 m freestyle swimming in 280 adolescent swimmers (r = −0.26, aged: 14.2 ± 1.7 years) [22,23]. The latter study also showed that faster swimmers had greater lean mass (48.06 ± 6.3 kg) compared to poorer performance swimmers (42.46 ± 4.9 kg) [22]. It appears that lean body mass is a significant contributing factor for performance outcome in sprint swimming. Thus, coaches may consider combining sprint swimming training with dry-land resistance training programs for potentially enhancing the lean body mass of their athletes. Indeed, studies have shown that dry-land training may be a useful tool for coaches to increase swimming performance in sprint swimmers [19,20,21,34]. Therefore, the results of the current correlation analysis between lean body mass and swimming performance reinforce the utility of dry-land resistance exercises to increase lean body mass in an attempt to subsequently enhance sprint swimming performance.

No significant change was found for CMJ after both priming programs. Therefore, DLP and SP failed to induce significant power enhancement in CMJ performance, a result that would indicate the possible neuromuscular mechanism for the decreased swimming time-trial. Similar to previous studies, CMJ power was significantly correlated with 50 m time performance after C, DLP, and SP [20,21,26]. Consequently, CMJ may be effectively used by coaches as an easy-to-use test tool for the evaluation of lower body power in an attempt to predict sprint swim performance in well-trained adolescent swimmers. In addition, RFD is a significant factor for performance in many power sports [35,36]. Previous studies have shown strong correlations between lean body mass and isometric force and RFD [36,37]. However, data are scarce regarding the link between sprint swim performance and fast force production. Results from previous studies in well-trained and international level swimmers showed that RFD was correlated with 50 m swim performance (r = −0.72) but not with 15 m sprint performance [20,26]. In the current study, IPT and RTD were significantly correlated with 50 m crawl sprint performance following C, DLP, and SP conditions. This finding further reinforces the recommendation that sprinter swimmers may incorporate dry-land resistance training in their training programs and focus on performing all repetitions with maximum intentional movement velocity [13,16,17]. Hence, during dry-land training, coaches should instruct athletes to perform repetitions with maximum intention of movement velocity regardless of the actual movement load.

Unfortunately, we were unable to evaluate possible neural, intramuscular, and hormonal responses after the priming programs which could lead to a better insight of the performance results. Moreover, the current study focused on 50 m crawl style sprint only; thus, the application of these results in other distances and swimming styles might be limited. In addition, since the results of the current study revealed significant correlation between lean body mass and performance, the age of the target group could be considered as a limiting factor, leading to the necessity to explore any possible effects in older swimmers. Although the findings of the current study are attractive, swimming coaches and strength and conditioning specialists should interpret the results with caution. More research has to be conducted to reach safe conclusions about the effect of priming training in swimming performance.

## 5. Conclusions

Swimming performance in 50 m crawl was significantly improved following DLP and SP compared to C, which underpins the effectiveness of the two different priming sessions applied 24 h before the time-trials. Thus, swimming coaches and strength and conditioning professionals may safely apply both priming programs to their athletes 24 h before a sprint event. Dry-land priming was performed with power-based exercises and with maximum intentional movement velocity especially during the concentric muscle contraction. Similarly, swimming priming should be performed with maximum swimming speed. Furthermore, lean body mass, CMJ power, IPT, and RTD were significantly linked with 50 m crawl swimming. Consequently, coaches should regularly monitor changes in lean body mass, power, and fast force production during training for potentially predicting changes in 50 m crawl swimming performance.

## Figures and Tables

**Figure 1 sports-10-00052-f001:**
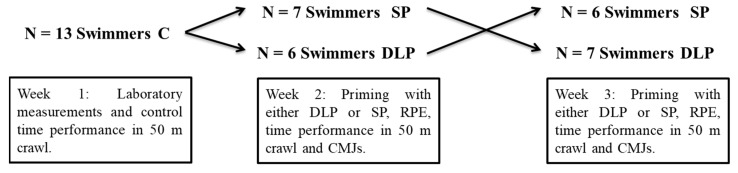
Counterbalanced study design leading to three experimental groups: control (C), dry land priming (DLP), and swimming priming (SP). RPE = rate of perceived exertion, CMJs = countermovement jumps.

**Figure 2 sports-10-00052-f002:**
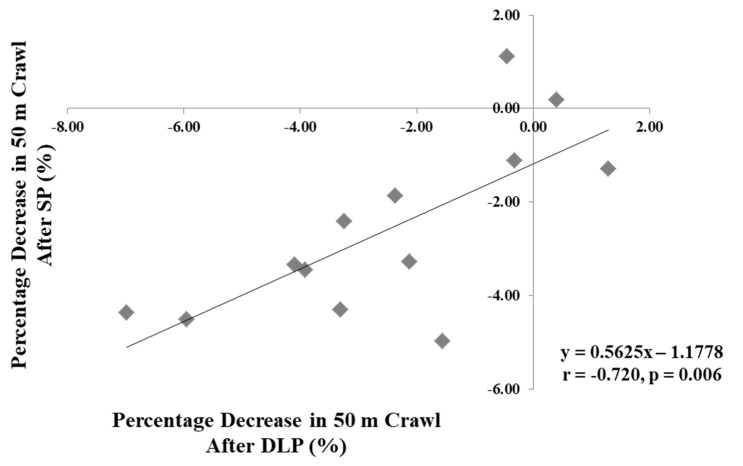
Correlation between the percentage decreases in 50 m crawl time-trial after dry-land priming (DLP) and the percentage decreases in 50 m crawl time-trial after swimming priming (SP).

**Table 1 sports-10-00052-t001:** Acute training programs for dry land and swimming priming.

Dry Land Priming	Slam balls 3 sets of 8 repetitions (4 kg for males, 2 kg for females). Countermovement jumps 3 sets of 8 repetitions. Stretch cords upper body swimming exercise imitating crawl swimming 3 sets of 12 repetitions for each side.	Rest between sets 2 min. Rest between repetitions for slam balls and CMJs 3 s. All repetitions performed with maximum voluntary velocity of movement.
Swimming Priming	Warm-up routine (700 m). Four sets of 50 m crawl swim starting from blocks followed by active swimming rest of 50 m back to start.	All swimming sprints performed with maximum voluntary velocity of swim. Ratio between sprint and rest was 1:4.

CMJs = countermovement jumps.

**Table 2 sports-10-00052-t002:** Results from laboratory measurements, control, dry-land, and swimming priming.

	Control	Dry-Land Priming	Swimming Priming
Body mass (kg)	62.4 ± 8.1	62.6 ± 8.2	62.5 ±8.0
50 m crawl time performance (s)	30.02 ± 2.73	29.24 ± 2.46 *	29.22 ± 2.48 *
CMJ height (cm)	32.3 ± 5.2	32.5 ± 5.4	32.3 ± 5.3
CMJ power (W)	2719.5 ± 537.7	2724.9 ± 576.7	2737.7 ± 526.1
CMJ (W/kg)	43.3 ± 4.7	43.5 ± 4.9	43.4 ± 4.7
Body fat (%)	17.4 ± 5.6		
Total lean mass (kg)	51.5 ± 6.8		
Trunk lean mass (kg)	26.9 ± 3.2		
Legs lean mass (kg)	16.9 ± 2.5		
Arms lean mass (kg)	5.0 ± 1.1		
IPT (Nm)	213.0 ± 46.5		
RTD20msec (Nm·s^−1^)	1419.2 ± 352.7		
RTD40msec (Nm·s^−1^)	1245.0 ± 270.1		
RTD60msec (Nm·s^−1^)	1185.5 ± 295.0		
RTD80msec (Nm·s^−1^)	1155.9 ± 272.0		
RTD100msec (Nm·s^−1^)	1130.8 ± 246.9		
RTD120msec (Nm·s^−1^)	1080.9 ± 217.2		
RTD150msec (Nm·s^−1^)	1000.5 ± 196.2		
RTD200msec (Nm·s^−1^)	873.4 ± 166.3		
RTD250msec (Nm·s^−1^)	746.4 ± 140.9		
RTD300msec (Nm·s^−1^)	632.0 ± 121.6		

* *p* < 0.05, significant difference from control, CMJ = countermovement jump, IPT = isometric peak torque, RTD = rate of torque development.

**Table 3 sports-10-00052-t003:** Correlation coefficients between body composition variables and 50 m crawl time performance after control, dry-land priming, and swimming priming.

	Body Fat	Total Lean Mass	Trunk Lean Mass	Legs Lean Mass	Arms Lean Mass
C	−0.285	−0.744 **	−0.748 **	−0.710 *	−0.721 *
DLP	−0.340	−0.785 **	−0.791 **	−0.758 **	−0.743 **
SP	−0.263	−0.739 **	−0.727 *	−0.740 **	−0.718 *

* *p* < 0.05, ** *p* < 0.01, C = control, DLP = dry land priming, SP = swimming priming.

**Table 4 sports-10-00052-t004:** Correlation coefficients between isometric peak torque and rate of torque development with 50 m crawl time performance after control, dry-land priming, and swimming priming.

	IPT	RTD	RTD	RTD	RTD	RTD	RTD	RTD	RTD	RTD	RTD
		20 ms	40 ms	60 ms	80 ms	100 ms	120 ms	150 ms	200 ms	250 ms	300 ms
C	−0.774 **	−0.370	−0.617 *	−0.596	−0.663 *	−0.647 *	−0.681 *	−0.767 **	−0.821 **	−0.793 **	−0.697 *
DLP	−0.746 **	−0.384	−0.625 *	−0.611 *	−0.689 *	−0.710 *	−0.731 *	−0.765 **	−0.812 **	−0.763 **	−0.683 *
SP	−0.813 **	−0.333	−0.555	−0.549	−0.611 *	−0.589	−0.625 *	−0.723 *	−0.811 **	−0.834 **	−0.771 **

* *p* < 0.05, ** *p* < 0.01, C = control, DLP = dry-land priming, SP = swimming priming, IPT = isometric peak torque, RTD = rate of torque development.

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
