# Peer review of "Comparison between Dry-Land and Swimming Priming on 50 m Crawl Performance in Well-Trained Adolescent Swimmers"

_sports, 2022, doi:10.3390/sports10040052_

Round 1

Reviewer 1 Report

In the present study, entitled "Effects of Dry-Land vs. Swimming Priming on 50m Crawl Performance in Well-Trained Adolescent Swimmers", the effects of two interventions were investigated and compared with each other and with a control group. The main issue was the design of the study. The maximum 50m time-trial crawl swim, which was considered as the control condition (C), should be in the counterbalanced design. The improvement in performance could be the result of daily training during the three-week study. Otherwise, change the title to "Comparison of two interventions...."

ln, 19-20. The improvement of performance in 50m crawl can be caused by normal physiological adjustments, because of the three-week interventions.

ln, 99. "followed 5-6 training sessions per week" Do these numbers correspond only to water training or even land training?

ln, 142-143. Thirty minutes cool down can blur their judgment.

Table 1. Countermovement jumps 3 sets of 8 repetitions. Move this sentence to the second position. Alternatively, merge the sentences as "Slam balls 3 sets of 8 repetitions (4kg for males, 2 kg for females) with 8 repetitions of countermovement jumps in-between".

ln, 183-184. Was the squat jump depth self-selected?

RESULTS

Figure 2. It is not necessary if the results are presented in Table 2.

Figure 3. Correlation is not necessary due to the absence of statistical differences in swimming performance after the two interventions.

DISCUSSION

ln, 290-299. This paragraph is redundant 

Author Response

Response

We would like to thank the reviewer for the time spent on this manuscript and the valuable comments and suggestions. We have now revised our manuscript according to reviewer’s suggestions.

In the present study, entitled "Effects of Dry-Land vs. Swimming Priming on 50m Crawl Performance in Well-Trained Adolescent Swimmers", the effects of two interventions were investigated and compared with each other and with a control group. The main issue was the design of the study. The maximum 50m time-trial crawl swim, which was considered as the control condition (C), should be in the counterbalanced design. The improvement in performance could be the result of daily training during the three-week study. Otherwise, change the title to "Comparison of two interventions...."

Response:

We thank the reviewer for point this to our attention. The counterbalanced design used in the current study has previously been applied to numerous of papers (J Strength Cond Res 28(12): 3484–3495, 2014; Journal of Strength and Conditioning Research, DOI: 10.1519/jsc.0000000000003792; International Journal of Sports Physiology and Performance, https://doi.org/10.1123/ijspp.2020-0094). However, we changed the title according to the reviewer’s suggestion.

ln, 19-20. The improvement of performance in 50m crawl can be caused by normal physiological adjustments, because of the three-week interventions.

Response:

We thank the reviewer for the intriguing comment. We agree with the reviewer. However, due to the obligations of the participants (school, studying and other extracurricular activities) it was not possible to perform the measurements with two or three days wash-out period. Regardless of this limitation, we have read a study which used one week wash-out period between conditions (J Strength Cond Res 32(3): 643–650, 2018) in well trained power athletes similar to our study design. We also believe that the counterbalanced design applied in the current study might have limited the training-induced effects caused by the systematic training.

ln, 99. "followed 5-6 training sessions per week" Do these numbers correspond only to water training or even land training?

Response:

We thank the reviewer for the question. This period corresponds in both water and dry land training. More specific athletes followed approximately two dry-land training sessions and 3-4 swimming training sessions. We have now added this inside the manuscript.

ln, 142-143. Thirty minutes cool down can blur their judgment.

Response:

We thank the reviewer for the comment. We followed the suggestion form Foster et al., 2001 (J. Strength Cond. Res. 15(1):109–115) and we have used this RPE response time in previous published studies of our laboratory (Applied Sciences, 2021, 11, 45. https://dx.doi.org/10.3390/app11010045; Journal of Human Kinetics volume 81/2022, 189-198, DOI: 10.2478/hukin-2022-0016). It also has been used to similar studies where RPE was the main measurement (J. Strength Cond. Res. 2004, 18, 353–358; J. Strength Cond. Res. 2004, 18, 796–802).

Table 1. Countermovement jumps 3 sets of 8 repetitions. Move this sentence to the second position. Alternatively, merge the sentences as "Slam balls 3 sets of 8 repetitions (4kg for males, 2 kg for females) with 8 repetitions of countermovement jumps in-between".

Response:

We thank the reviewer for the suggestion. We moved the sentence to the second position.

ln, 183-184. Was the squat jump depth self-selected?

Response:

We thank the reviewer for the interesting question. Yes, the depth during the countermovement jumps both in training and in measurement was self-selected.

RESULTS

Figure 2. It is not necessary if the results are presented in Table 2.

Response:

We agree with the reviewer. We deleted figure 2.

Figure 3. Correlation is not necessary due to the absence of statistical differences in swimming performance after the two interventions.

Response:

We thank the reviewer for the suggestion. We understand the sceptical of the reviewer and indeed there was no significant difference between the two training programs. However, we feel that this strong correlation between the percentage decreases in 50m time-trials significant contributes building our theory. From the one hand it shows that all athletes experienced almost identical decreases in performance following both priming programs while on the other hand it shows that both DLP and SP may lead to similar performance enhancements which strengthens the ANOVA results. Also, readers may observe the individual training-induced adaptations to each priming program. Thus, we would like to keep this correlational figure. We hope that our response will not upset the reviewer.

DISCUSSION

ln, 290-299. This paragraph is redundant

Response:

Thank you for the comment. We have provided a detailed response above.

Reviewer 2 Report

This is an interesting study from the practical point of view, because it gives an insight to perspectives in swimming training.

Having reviewed the present paper I have some comments that I would like to share with the Authors.

Methods

More information about study design should be provided. As the number of participants is relatively small, the way of sample size calculation should be described. If the sample size was not calculated, the reason of engaging such a number of participants needs to be provided. Moreover, I suggest describing the way of randomization the participants (group DLP vs group SP). Are DLP and SP protocols Authors' invention or these protocols have been already used in previous studies? Please provide information about that.

Results

line 229: "No significant difference was observed between DLP and SP" - it is unclear which differences are discussed.

Figure 2 - please consider removing this figure, as it reflects the data presented in Table 2.

Discussion

line 284-286: I am not sure if reference no. 34  has much in common with the present paper, as psychological variables were not identified in your group of swimmers.

line 304-306: In fact, Latt et al. (reference no. 22) presented the correlation between 100-m time and fat-free mass, not lean body mass. Moreover, this correlation was insignificant (r=-0,506, p>0.05). Yu et al. (BMC Pharmacol Toxicol 2013, DOI: 10.1186/2050-6511-14-53) present the differences between fat-free mass and lean body mass. Please consider re-writing this sentence.

Author Response

This is an interesting study from the practical point of view, because it gives an insight to perspectives in swimming training.

Having reviewed the present paper I have some comments that I would like to share with the Authors.

Response

We would like to thank the reviewer for the time spent on this manuscript and the valuable comments and suggestions. We have now revised our manuscript according to reviewer’s suggestions.

Methods

More information about study design should be provided. As the number of participants is relatively small, the way of sample size calculation should be described. If the sample size was not calculated, the reason of engaging such a number of participants needs to be provided.

Response:

We thank the reviewer for the intriguing comment. Previous studies have used similar sample sizes to ours (International Journal of Sports Physiology and Performance, DOI: https://doi.org/10.1123/ijspp.2016-0276; International Journal of Sports Physiology and Performance, https://doi.org/10.1123/ijspp.2020-0094), while an earlier study of our laboratory with similar experimental and statistical design, used a G*Power analysis to calculate the number of participants (Journal of Strength and Conditioning Research, DOI: 10.1519/jsc.0000000000003792). We have now added the power analysis results in the statistical analysis paragraph.

Moreover, I suggest describing the way of randomization the participants (group DLP vs group SP). Are DLP and SP protocols Authors' invention or these protocols have been already used in previous studies? Please provide information about that.

Response:

We thank the reviewer for the comment and question. Athletes were randomly assigned into DLP and SP protocols. We mention this in line 113.

In addition, the training protocols are similar to other studies in swimmers and soccer players (low load - high intensity training programs; Int. J. Sports Physiol. Perform. 2017, 12, 605-611. doi.org/10.1123/ijspp.2016-0276; Int. J. Sports Physiol. Perform. 2021, 16, 407-414. doi.org/10.1123/ijspp.2020-0094). However, our aim was to investigate a ballistic – power based dry-land training program on swimming performance, while the swimming protocol might have been a little shorter compared to previous study which used 1200 meters of various swimming drills (Int. J. Sports Physiol. Perform. 2017, 12, 605-611. doi.org/10.1123/ijspp.2016-0276), compared to our study where we used approximately 700 meters for warm up and 400 meters as the main training intervention.

Results

line 229: "No significant difference was observed between DLP and SP" - it is unclear which differences are discussed.

Response:

We thank the reviewer for the comment. Differences are referred to time-trial performance. We have now added this inside the manuscript.

Figure 2 - please consider removing this figure, as it reflects the data presented in Table 2.

Response:

We thank the reviewer for the suggestion. We have now deleted figure 2.

Discussion

line 284-286: I am not sure if reference no. 34  has much in common with the present paper, as psychological variables were not identified in your group of swimmers.

Response:

We thank the reviewer for the comment. We have now deleted the reference 34.

line 304-306: In fact, Latt et al. (reference no. 22) presented the correlation between 100-m time and fat-free mass, not lean body mass. Moreover, this correlation was insignificant (r=-0,506, p>0.05). Yu et al. (BMC Pharmacol Toxicol 2013, DOI: 10.1186/2050-6511-14-53) present the differences between fat-free mass and lean body mass. Please consider re-writing this sentence.

Response:

We thank the reviewer for pointing this to our attention. We apologise for this. We have now replaced this reference with Stager, J.M.; Cordain, L.; Becker, T.J. Relationship of body composition to swimming performance in female swimmers. J. Swimming Research 1984, 1, 21-6.